# The genomic history of the indigenous people of the Canary Islands

Javier G. Serrano [1], Alejandra C. Ordóñez [2], Jonathan Santana [2], Elías Sánchez-Cañadillas [2], Matilde Arnay[3], Amelia Rodríguez-Rodríguez [2], Jacob Morales [2], Javier Velasco-Vázquez[4], Verónica Alberto-Barroso [5], Teresa Delgado-Darias[6], M. Carmen Cruz de Mercadal[6], Juan Carlos Hernández[7], Marco A. Moreno-Benítez[5], Jorge Pais[8], Harald Ringbauer [9], Martin Sikora [10], Hugh McColl [10], Maria Pino-Yanes [11,12], Mariano Hernández Ferrer [13], Carlos D. Bustamante[14] & Rosa Fregel [1,14] ✉

The indigenous population of the Canary Islands, which colonized the archipelago around the 3[rd] century CE, provides both a window into the past of North Africa and a unique model to explore the effects of insularity. We generate genome-wide data from 40 individuals from the seven islands, dated between the 3[rd]–16[rd] centuries CE. Along with components already present in Moroccan Neolithic populations, the Canarian natives show signatures related to Bronze Age expansions in Eurasia and trans-Saharan migrations. The lack of gene flow between islands and constant or decreasing effective population sizes suggest that populations were isolated. While some island populations maintained relatively high genetic diversity, with the only detected bottleneck coinciding with the colonization time, other islands with fewer natural resources show the effects of insularity and isolation. Finally, consistent genetic differentiation between eastern and western islands points to a more complex colonization process than previously thought.

North Africa has a unique geographical situation that has favored demic diffusion between continents. The Sinai Peninsula is a land bridge that supports migratory routes between the African continent and Eurasia. At the north, the Mediterranean Sea has been the center of the cultural and economic trade that shaped the history of the surrounding human populations. Due to the effect of the warm and humid climate on human remains, ancient DNA (aDNA) from the North African region has remained largely understudied; only three prehistoric populations from the Upper Paleolithic to the Late Neolithic of the Western North African region have been reported so far (Fig. 1a).

[1]Evolution, Paleogenomics and Population Genetics Group, Department of Biochemistry, Microbiology, Cell Biology and Genetics, Universidad de La Laguna, San Cristóbal de La Laguna, Santa Cruz de Tenerife, Spain. [2]Tarha Group, Department of Historical Sciences, Universidad de Las Palmas de Gran Canaria, Las Palmas de Gran Canaria, Las Palmas, Spain. [3]Bioanthropology: Paleopathology, Diet and Nutrition in Ancient Populations Group, Department of Prehistory, Anthropology and Ancient History, Universidad de La Laguna, San Cristóbal de La Laguna, Santa Cruz de Tenerife, Spain. [4]Servicio de Patrimonio Histórico, Cabildo de Gran Canaria, Las Palmas de Gran Canaria, Las Palmas, Spain. [5]Tibicena Arqueología y Patrimonio, Las Palmas de Gran Canaria, Las Palmas, Spain. [6]El Museo Canario, Las Palmas de Gran Canaria, Las Palmas, Spain. [7]Museo Arqueológico de La Gomera, San Sebastián de La Gomera, Santa Cruz de Tenerife, Spain. [8]Museo Arqueológico Benahoarita, Los Llanos de Aridane, Santa Cruz de Tenerife, Spain. [9]Department of Archaeogenetics, Max Planck Institute for Evolutionary Anthropology, Leipzig, Germany. [10]Lundbeck Foundation GeoGenetics Centre, Globe Institute, University of Copenhagen, Copenhagen, Denmark. [11]Genomics and Health Group, Department of Biochemistry, Microbiology, Cell Biology and Genetics, Universidad de La Laguna, San Cristóbal de La Laguna, Santa Cruz de Tenerife, Spain. [12]CIBER de Enfermedades Respiratorias, Instituto de Salud Carlos III, Madrid, Spain. [13]Molecular Genetics and Biodiversity Group, Department of Biochemistry, Microbiology, Cell Biology and Genetics, Universidad de La Laguna, San Cristóbal de La Laguna, Santa Cruz de Tenerife, Spain. [14]Department of Genetics, Stanford University, Stanford, CA, USA. ✉e-mail: rfregel@ull.edu.es

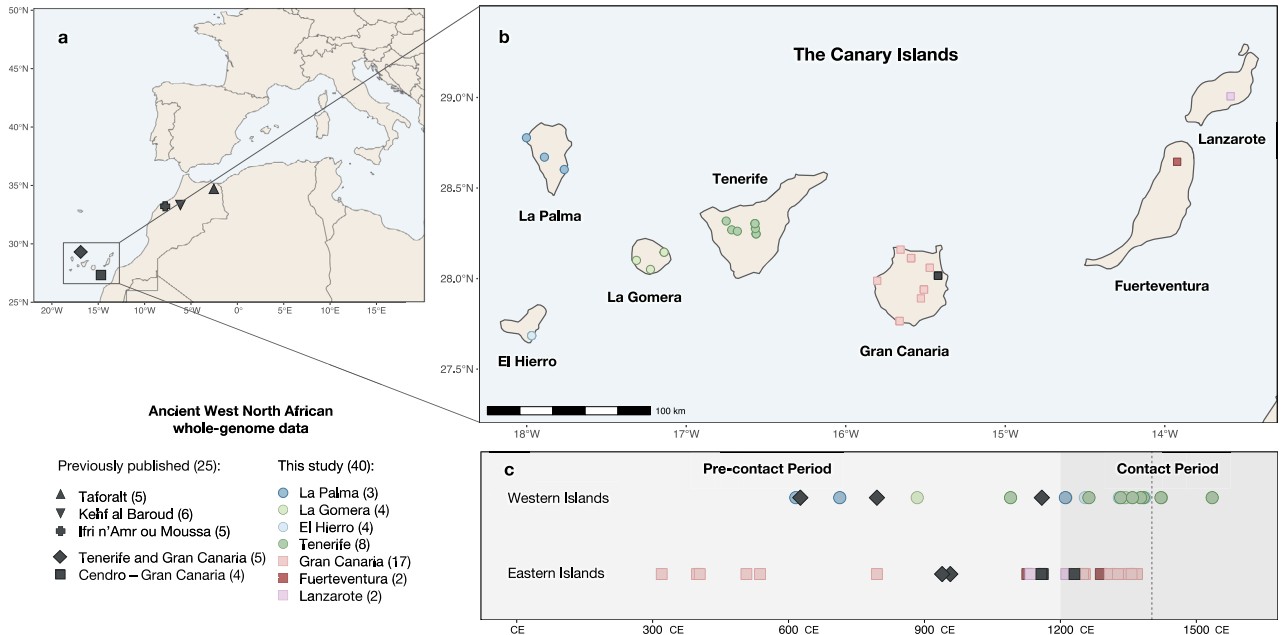

**Fig. 1 | Geographical and temporal adscription of the Canary Islands indigenous individuals. a** Available ancient whole-genome data from western North Africa obtained from the literature: Taforalt[2], Kehf al Baroud and Ifri n'Amr ou Moussa[5]; decontextualized individuals from Tenerife and Gran Canaria[15], and individuals from Cendro site in Gran Canaria[21]. **b** Geographical adscription of the archaeological sites considered in this study. Individuals from the Canary Islands with no archaeological site adscription are not included in **b**: five previously published individuals from Gran Canaria and Tenerife[15] and an individual from Fuerteventura generated in this study. **c** Available radiocarbon data for the Canary Islands indigenous genomes. The dotted line in **c** indicates the start of the Castilian conquest (1402). The darker grey tile indicates the period in which the indigenous people of the Canary Islands were in contact with European seafarers. Previously published genomes are indicated in grey while genomes generated in this study are indicated in other colors. Maps made with Natural Earth (https://naturalearthdata.com).

The current North African genomic pool has been shaped by genetic influxes from sub-Saharan Africa, Europe, the Middle East, and the Caucasus into an autochthonous ancestral population[1]. This autochthonous component descends from a population linked to the Upper Paleolithic population from Taforalt (present-day Morocco), dated to around 15,000 years before present (BP). Their genome-wide ancestry is consistent with a substantial Eurasian origin, suggesting a Paleolithic back migration to Africa from Eurasia as proposed before[2–4]. Later Early Neolithic genomes (7000 BP) were genetically similar to the Taforalt population, showing that the first stages of the Neolithic revolution in North Africa were driven by the acquisition of farming techniques by the local population and not by a population turnover[5]. However, the later phase of the Neolithic was characterized by the movement of people, as Late Neolithic genomes (5000 BP) showed admixture between the local populations and early European farmers[5]. Nevertheless, to our knowledge, no other genomic information has been obtained for the Western North African region until the late Medieval Period. From the 7th century after the Common Era (CE), the Islamic invasions from the Arabic peninsula changed the cultural and genetic background of most of the local populations[6], making it difficult to disentangle the genomic history of the region from the Late Neolithic to the Antiquity.

A unique perspective window into the past of North Africa can be accessed through the study of the indigenous people from the Canary Islands (CIP). The Canarian archipelago consists of seven main volcanic islands in the Atlantic Ocean, 100 km off the northwest African coast (Fig. 1b). This population of North African origin was most probably isolated from the mainland before the arrival of the Islamic invasions, thus representing a genetic reservoir of the western North-African ancestry before the impact of the Arab conquest. Therefore, genomic data from the CIP is key to understanding the genetic history of the understudied region of Western North Africa.

Current radiocarbon dating evidence suggests that the Canaries were first populated between the second and fifth centuries CE[7]. Archaeological evidence indicates that later connections between the islands and the African coast were very limited, and that the islands remained practically isolated until contact with European seafarers and explorers in the 14th century[8], who eventually conquered the archipelago in the 15th century and admixed with the surviving indigenous people. Previous genetic data pointed to a clear North African origin for the Canarian indigenous population, using uniparental markers[9–14] and genome-wide data[15,16]. Spatial differences have been observed in the islands' populations regarding both the diversity and composition of the mitochondrial DNA (mtDNA), with islands with more natural resources having higher genetic diversity[9].

Given the tremendous biogeographic diversity that characterizes the archipelago, the first settlers were driven to develop different life strategies that resulted in different adaptive processes in terms of social complexity, subsistence practices, and demographic development, making the settlement of the Canaries an intriguing human colonization process[17–20]. In that sense, the Canary Islands can also be used as unique laboratories to study complex demographic processes from a genetic perspective, including colonization, isolation, or admixture with other populations. On that basis, we perform a study of the whole Canary Islands indigenous population at a genome-wide level to obtain information on the prehistory of North Africa and understand how isolation and insularity shaped their genetic structure.

Previously published genome-wide data from the CIP included five decontextualized individuals from the islands of Tenerife and Gran Canaria, ranging from the 7th to the 15th centuries CE[15], from a 19th century private collection conserved at the Anatomical Museum Edinburgh University with no archaeological information; and four from the Cendro site in Gran Canaria dated around the 12th century

CE[21]. Here, we generated 9 medium to low-coverage genomes (5.82X−0.36X) by shotgun sequencing and genome-wide data from another 31 individuals by in-solution capture from the CIP (Supplementary Data 1). The individuals are distributed over twenty-three archaeological sites from the seven main islands (Fig. 1b; Supplementary Data 1), comprising a time transect of c. 1300 years from the Canarian indigenous history, from the 3rd to the 16th century CE (Fig. 1c; Supplementary Data 1). Measures to avoid and monitor contamination from modern DNA were applied during sample manipulation. Ancient DNA was extracted from teeth or bone[22], built into double-stranded indexed libraries[23], and sequenced on an Illumina NextSeq 500. To overcome limitations due to DNA degradation, we applied two different capture methods to enrich on human reads: one targeting the whole genome and one targeting variable sites (see Supplementary Note 2).

## Results

### Ancestry inference of the Canary Islands indigenous population

Most male individuals from the CIP are classified within the basal Y-chromosome E-M183* North African lineage, whose emergence (2000–3000 years ago)[24] precedes the time of the Canary Islands colonization. Other haplogroups observed are E-M33, T-M184, R-M269, and E-M78, which can be associated to sub-Saharan African, European Neolithic, and Bronze Age expansions in North Africa (Supplementary Note 4; Supplementary Data 2).

To assess genetic variation, we first computed a principal components analysis (PCA) using present-day data and then projected ancient individuals. When projecting using the Human Origins dataset[25] as the reference panel, the CIP are placed with Eurasian populations (Fig. 2a), and form a cluster that is close to Late Neolithic Moroccans and modern North Africans, as previously observed[5,15]. In PC1, they cluster closer to sub-Saharan populations than Late Neolithic Moroccans, consistent with the presence of sub-Saharan African mtDNA[9] and Y-chromosome lineages (Supplementary Note 4). Compared with Late Neolithic Moroccans, the indigenous people are shifted towards the European Middle/Late Neolithic and Bronze Age people, while Late Neolithic North Africans are shifted towards the Early Neolithic individuals from Anatolia and Europe.

To estimate the genetic contribution of the ancestral populations to the CIP, we performed unsupervised clustering analyses using ADMIXTURE[26] (Fig. 2b). The CIP appears to be composed by the admixture of three components (K = 8): an ancient Maghrebi and an early Neolithic European contribution, as well as a component associated to steppe populations, that, in contrast to the other two components, is absent in Late Neolithic Moroccans. As the first permanent settlement of the Canary Islands has been estimated to have happened around the third century CE[17], additional migration waves are expected to have reached North Africa by that time. Indeed, given the presence of Bell-Beaker pottery in the North African archaeological record[27], this steppe component could be related to the expansion of European Bronze Age populations into North Africa[5,16]. Moreover, this observed steppe component could also be connected to the spread of the Mediterranean cultures into western North Africa, such as Punics and Romans since the 9th century BC and 2nd century BC, respectively[28].

To properly model CIP, we performed admixture modelling analyses using qpAdm[29]. Our results indicate that the indigenous population is best modeled involving the contribution of Late Neolithic Morocco (modelled by Morocco_LN), a source of Paleolithic/Early Neolithic Maghrebi ancestry (Morocco_IB or Morocco_EN, respectively), a source of steppe ancestry (either Bell Beakers from Germany [Germany_BB], Yamnaya from Russia [Russia_EBA_Yamnaya] or hunter-gatherers from Rusia [Russia_HG_Karelia]), and Mota as a proxy source of sub-Saharan ancestry (Supplementary Data 3). Therefore,

considering the best-fitting model (Supplementary Data 3; Fig. 3b), the CIP ancestry can be explained as the admixture of Morocco_LN (73.3% ± 2.2%), Morocco_EN (6.9% ± 1.0%), Germany_BB (13.4% ± 1.8%) and Mota (6.4% ± 1.3%). We also tested if the steppe ancestry reached the islands via Romanized Berber populations. Models only worked when involving Roman populations with either North African (Punics from Ibiza and Sardinia) or Middle Eastern (Romans from England) influence[30–32] (Supplementary Data 4).

Given the time of the human colonization of the islands, it is also interesting to compare them to modern North Africans to get an insight into the impact that later migrations left on this region. Modern North Africans exhibit a lower proportion of the ancient North African component when compared to the Canarian natives in an unsupervised admixture analysis, decreasing from the west towards the eastern areas (Fig. 2b). As expected from historical records, North Africans have differences correlating to additional migration waves reaching this region in the last two millennia, including a higher Near Eastern contribution likely due to the Arab expansion (shown in grey) and a higher Sub-Saharan African one due to trans-Saharan migrations (red), most probably related to the slave trade and commercial interactions[6].

In order to understand the impact of the European conquest, it is also compelling to compare the genetic composition of the indigenous and the modern people of the Canary Islands. For this purpose, using qpAdm, we modeled the modern Canarians considering Spanish, Yoruba from Guinea Gulf region, and CIP as source populations. Our results indicate that modern Canarians can be modeled as the result of the admixture of 79.7% ± 1.0% contribution from Spain, 17.8% ± 1.3% from the indigenous people, and 2.6% ± 0.5% from sub-Saharan Africa (Supplementary Note 7). This result demonstrates the significant impact of conquest and colonization on the indigenous people. Although the sub-Saharan contribution is low, it is evidence of the footprint of the transatlantic slave trade in the islands confirming mitochondrial DNA evidence[16].

### Population structure within the Canarian archipelago

Our large dataset allowed us to investigate between-island population structure of the CIP. PCA analysis identified two consistently differentiated clusters, with all western islands (El Hierro, La Palma, La Gomera, and Tenerife) placed closer to Upper Paleolithic and Early Neolithic North Africans, and all eastern islands (Gran Canaria, Fuerteventura, and Lanzarote) cluster closer to ancient and modern Europeans. The only exceptions are one individual from Tenerife (CAN.039) from the period of contact with European explorers and one indigenous individual from La Palma (CAN.035).

ADMIXTURE analysis also identifies differences between regions, with the western islands showing a higher proportion of the autochthonous North African component and a lower contribution associated with steppe populations than eastern islands (Fig. 2a). These results correlate with the one observed in the PCA. They can explain the clustering of eastern and western people of the CIP, with the first group having a greater steppe contribution and the other being more akin to ancient North Africans. Indeed, when we perform outgroup $f3$-statistics considering the amount of shared ancestry with populations in the Human Origins dataset for the total and the insular populations, we observe that shared drift between eastern Canarian people and Neolithic and Bronze Age populations from Europe is higher than those inferred for the western individuals (Supplementary Note 5.3).

When formal admixture modeling is performed for both regions (Fig. 2c, Supplementary Data 5), western islands have a higher contribution of Morocco_EN (8.3% ± 1.1%) than eastern islands (4.9% ± 1.1%). Although the values overlap, the Germany_BB component is also slightly different between regions, with a lower value in the western (11.4% ± 1.9%) than in the eastern islands (16.0% ± 2.0%). When we analyze each island by itself, El Hierro and Tenerife have the highest

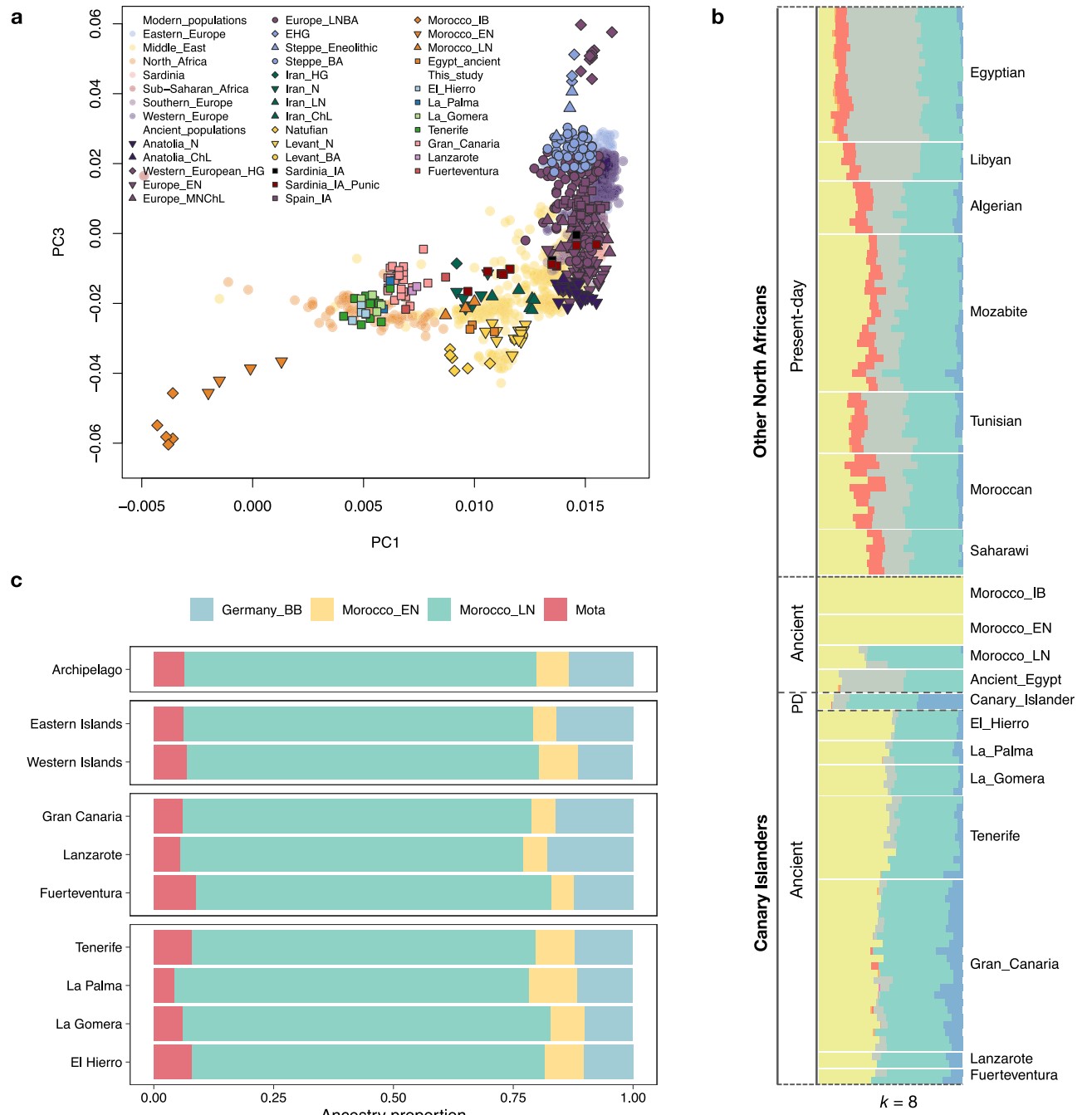

**Fig. 2 | Genetic composition of the Canary Islands indigenous people and their relationship to other ancient and modern populations. a** Principal Component Analysis (PCA) of the indigenous population of the Canary Islands in relation to other ancient and modern individuals from Europe, the Middle East and Africa, where the differentiation between western (green/blue) and eastern islands (red/violet) can be observed. **b** Unsupervised clustering analysis of the ancient and current populations from North Africa and the Canary Islands, both using the Human Origins panel, and based on ADMIXTURE results for $K = 8$. PD: present-day. **c** Ancestry proportions obtained for the qpAdm modeling of the CIP for the best-fitting model (Morocco_EN, Morocco_LN, Germany_BellBeaker and Mota), considering the entire archipelago, western and eastern islands and insular populations.

Morocco_EN contribution (8.2% ± 1.5% and 8.2% ± 1.2%, respectively). In comparison, Gran Canaria and Lanzarote have the highest Germany_BB contribution (16.2% ± 2.2% and 17.9% ± 3.3%, respectively). At an individual level, admixture values are homogenous within islands, even for individuals belonging to different time periods (Supplementary Note 7, Supplementary Data 6).

Differences are also detected regarding insular genetic diversity. When we estimate the genome-wide diversity of the islands' population using popstats[33] and the MEGA-Human Genome Diversity Project (HGDP)[34,35] dataset, Gran Canaria, Tenerife, and La Palma show the

highest heterozygosity estimations (0.215 ± 0.008, 0.213 ± 0.006, 0.215 ± 0.000, respectively), close to Late Neolithic populations from Morocco (Supplementary Note 8.2 and Supplementary Data 7). On the contrary, Fuerteventura shows the lowest heterozygosity estimates (0.184 ± 0.003), followed by Lanzarote and El_Hierro (0.193 ± 0.002 and 0.194 ± 0.01, respectively). La Gomera shows intermediate values (0.195 ± 0.005). In order to account for sample bias, heterozygosity estimations of the better characterized island populations were performed for pair of individuals, mimicking the sample size of the islands of Lanzarote and Fuerteventura. Although some variation was

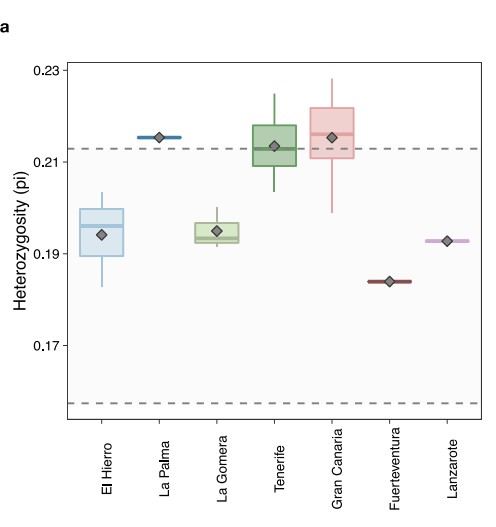

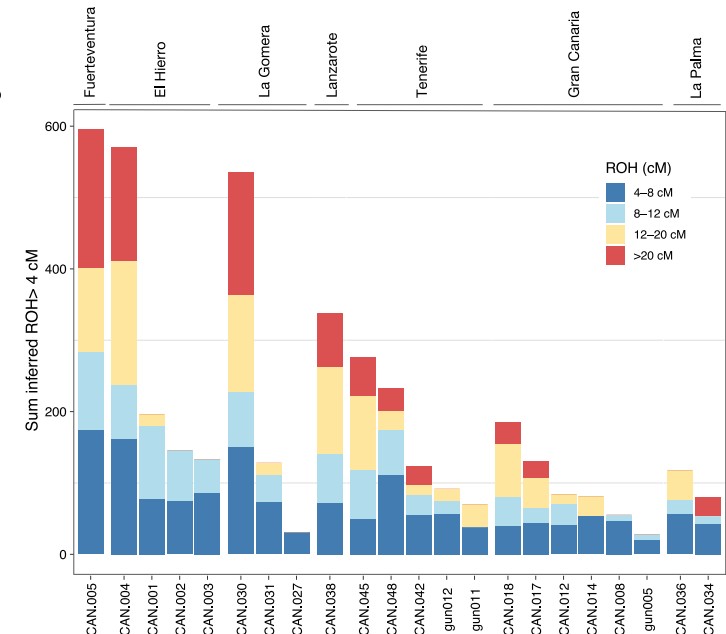

**Fig. 3 | Observed heterozygosity and runs of homozygosity for the Canary Islands indigenous people. a** Heterozygosity estimations obtained for all the islands (all individuals considered: El Hierro $n = 4$, La Palma $n = 3$, La Gomera $n = 4$, Tenerife $n = 11$, Gran Canaria $n = 23$, Fuerteventura $n = 2$, Lanzarote $n = 2$). The dotted lines represent the *pi* estimation obtained from Morocco_EN ($pi = 0.157$) and Morocco_LN ($pi = 0.213$). **b** Individual ROH results for the indigenous people of the Canary Islands.

observed in the heterozygosity estimations (see Supplementary Note 8.2), mean and median values for the islands of Tenerife and Gran Canaria are higher than those observed for El Hierro and La Gomera (Fig. 3a), and none of the pairs from Tenerife and Gran Canaria produced values as low as those observed for Lanzarote ($pi = 0.193$) and Fuer- ententura ($pi = 0.184$) (Supplementary Note 8.2). When looking at the archaeological sites with at least two individuals (Supplementary Note 8.2; Supplementary Data 7), El Hierro's Punta Azul site presents the lowest heterozygosity value ($0.196 \pm 0.001$) as inferred using mtDNA[9,14]. Interestingly, our analyses show that insular populations were differentiated both on their genetic composition and genetic diversity values.

## Genomic history of ancient Canarians

To investigate parental relatedness and consanguinity in the Canarian indigenous population, we also performed an estimation of the presence of runs of homozygosity (ROHs), using hapROH[36] and the MEGA-HGDP dataset (Fig. 3b and Supplementary Data 8). ROHs longer than 4 cM are detected in all the analyzed Canarian individuals ($n = 22$). Based on our results, most indigenous individuals have 20–60 cM of their genome composed of shorter ROH segments of 4–8 cM ($ROH_{[4,8]}$). This relatively high level of parental relatedness is a sign of recent background ROH, indicative of low population sizes[36]. We also examined individuals having >20 cM ROH ($sROH_{>20cM}$) segments greater than 50 cM, which is a signature of very small and isolated populations and consanguinity[36]. We identified five ancient individuals exceeding this long ROH threshold with total ROHs larger than 100 cM (Supplementary Data 8): four individuals from the smallest islands and/or islands with more ecological constraints (La Gomera, El Hierro, and Fuerteventura) along with an individual from Tenerife. All of them are from the latest stages of the indigenous period of the Canaries (from the 12th century onwards), which can be indicative of the effect of isolation in the later centuries. Remarkably, $sROH_{>20cM}$ above 50 cM are very uncommon in the global aDNA record: only 3% of the ancient individuals analyzed met this feature[36] and, notably, 20% of them are located on islands, such as Malta[37]. Although the first European written

records reported strict rules against marriage among members of the same group in the indigenous populations[38], our results indicate sporadic close-kin mattings, probably due to small population sizes and cryptic relatedness.

Considering the elevated background relatedness in the indigenous people, we estimated the effective population size ($N_e$) of the Canarian populations using hapROH (Fig. 4a). For the whole population of the archipelago, we observe that the $N_e$ was maintained at 411 individuals (95% CI: [371–460]), a slightly lower value than previously reported considering only ancient genomes from Gran Canaria and Tenerife[39]. In view of the indigenous population of each island, we observe that Fuerteventura ($N_e = 75$; 95% CI: [56–101]) and Lanzarote ($N_e = 151$; 95% CI: [106–221]) have the populations with the smallest $N_e$, followed by the population from El Hierro ($N_e = 205$; 95% CI: [163–263]). However, we must consider that we only sampled two individuals from Lanzarote and Fuerteventura and the actual $N_e$ values for these islands could be higher as these individuals might not be representative (see Supplementary Note 9.1). Furthermore, people from Gran Canaria had the highest $N_e$ estimations ($N_e = 460$; 95% CI: [470–697]), followed by the population from La Palma ($N_e = 395$; 95% CI: [312–611]) and Tenerife ($N_e = 285$; 95% CI: [323–490]), agreeing with our heterozygosity estimations. Surprisingly, La Gomera has the second-largest $N_e$ estimation of all the islands ($N_e = 429$; 95% CI: [292–664]). However, for La Gomera, we have an individual from an earlier occupation phase of the archipelago (CAN.027; 9th century) and two from a later period (12th–14th centuries), while individuals from Fuerteventura, Lanzarote, and El Hierro are all from the late occupation phase (12th century onwards). If effective population size had decreased over time, this difference might affect $N_e$ estimations. When we removed CAN.027 from the analysis, we obtained a more reduced $N_e$ for La Gomera ($N_e = 281$; 95% CI: [180–473]), more in line with that observed in other islands with low heterozygosity values.

When exploring how $N_e$ evolved around the 1300-year-transect, we observe a decreasing $N_e$ value over time when the whole archipelago sample is included (Fig. 4a). However, when considering the origin of the individuals, this decrease of $N_e$ is only observed for certain

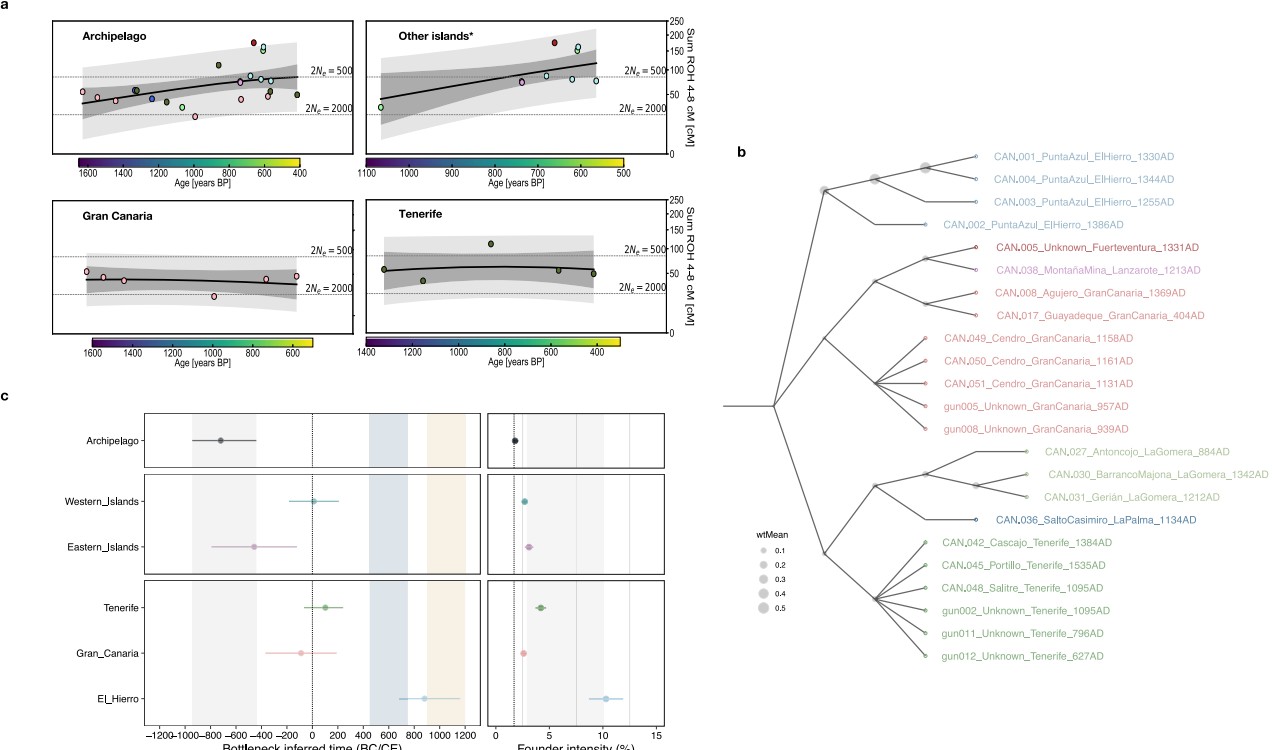

**Fig. 4 | Founder events, effective population size ($N_e$) trends and IBD clustering observed for the Canary Islands population.** In **a** $N_e$ trends are shown for the entire archipelago, smallest islands/islands with the least natural resources (La Gomera, El Hierro, Fuerteventura and Lanzarote); Gran Canaria, and Tenerife. *Excluding La Palma. The trend line represents the estimated mean of $N_e$, while its 95% empirical confidence intervals are shown in dark grey. The 95% empirical confidence intervals for individual $N_e$ are shown in light grey. **b** Clustering of individuals based on the inferred IBD segments in which each node represents an individual (labeled as individual_archaeological site_island_average RCD) colored based in the island they belong. Circles represent the proportion of shared IBD segments. **c** Founder event time range (left) and intensity (right) for the entire archipelago, western and eastern islands; and islands with $N \geq 4$ and coverage $\geq 0.1X$ are shown (Archipelago $n = 47$, Eastern Islands $n = 27$, Western Islands $n = 20$, El Hierro $n = 4$, Tenerife $n = 9$, Gran Canaria $n = 23$). Data are presented as mean values and 95% confidence intervals. For the founder effect time plot, the grey tile represents the time range of the putative archipelago's founder event, while the blue tile represents the Vandal Minimum period range, and the yellow tile represents the Medieval Warm Period. In the intensity founder effect plot, the grey tile represents the intensity range of most of the founder events occurred in continental populations, while the dotted line represents the intensity of the founder effect in Ashkenazi Jews (1.7%).

islands. Congruently with the ROH analysis, the reduction in $N_e$ is due to the inclusion of individuals from the late occupation period from the islands of El Hierro, La Gomera, and Lanzarote (Fig. 4a). If we consider the 9th-century individual from La Gomera (CAN.027) as representative of the inbreeding level of the population at that time, we detect a steep increase in inbreeding levels over time for the islands of La Gomera, Lanzarote, Fuerteventura, and El Hierro. On the contrary, when individuals from Gran Canaria and Tenerife (the two islands with larger sample sizes) are considered independently, the $N_e$ value remains stable for the entire indigenous occupation period.

To determine if differentiated migration patterns are responsible for differences in the effect of genetic drift over time, we inspected for shared genomic segments that are identical by descent (IBD) using IBDseq[40]. Genetic clustering of the CIP based on IBD-sharing shows three differentiated groups in the archipelago: one containing individuals from El Hierro, one with individuals from the remaining western islands, and the other with individuals from the eastern islands (Supplementary Note 12). Individuals from El Hierro form a separate cluster as they share a high fraction of their genome in IBD (Fig. 4b), probably due to the consanguinity derived from isolation and low effective population sizes. In the western cluster, La Gomera and La Palma individuals are separated from Tenerife, and each island is also placed in its own cluster. In the eastern cluster, the individual from Lanzarote is placed with the one from Fuerteventura and both are part of a separated cluster with CAN.008 and CAN.017 from Gran Canaria

(Fig. 4b). The remaining individuals from Gran Canaria are clustered together by site. Individuals CAN.049 to CAN.051 belong to the Cendro site and are all dated around the 12th century. Gun005 and gun008 (both dated around the 10th century) come from a private collection and were probably obtained from the same site, although it is impossible to know due to the loss of their archaeological context[15]. This analysis suggests the predominant driver in genetic relatedness is not extensive migrations between eastern and western islands and between islands within regions, as insular populations share more IBD with themselves than other islands.

To evaluate how population sizes changed in the past, we estimated the time and intensity of $N_e$ reductions in the islands' different populations (bottlenecks or founder events) using ASCEND[41] (Fig. 4c and Supplementary Data 10). First, we considered the whole archipelago population and observed a shared founder event that occurred around 720 BCE (95% CI: 944–440 BCE), coinciding with the 5th to the 11th centuries BCE. This population reduction occurred with a low-moderate intensity (95% CI: 1.7–2.0) (Fig. 4c), comparable with founder events seen in continental populations[41]. Island populations tend to experience more extreme founder events due to low founder population sizes. Given the intensity of the founder effect and the timing, we propose that this bottleneck could have affected the continental founder population time before their arrival to the archipelago. Western populations share a bottleneck/founder event that occurred around the end of the first century CE (95% CI: 184 BCE–208 CE), while

eastern individuals share an earlier event between the first and 7th centuries BCE (95% CI: 793 BCE–121 BCE) (Fig. 4c). Nevertheless, the intensity of the bottleneck events in both the eastern and western regions (on average 3.1% and 2.7%, respectively) are comparable to those expected for islands (Supplementary Note 9.2). The best approach to detect bottlenecks associated with an insular colonization event is to analyze island's populations separately, as each one experienced independent founder events. For Gran Canaria, we observe a $N_e$ reduction event coinciding with the 4th century BCE and the 2nd century CE, analogous to the potential eastern bottleneck in intensity (95% CI: 2.3%–2.9%) and time range. Westwards, the population from Tenerife could have experienced a more drastic event (95% CI: 3.7%–4.7%) than the whole western region but in the same period (95% CI: 66 BCE–242 CE) as the ones observed in other island populations[41]. Finally, we observed that El Hierro population went through a strong bottleneck event between the 7th and 12th centuries CE, a few generations before the individuals included in the study lived (Fig. 4c). The intensity of this event is ~10%, considerably greater than the rest of the events studied before[41]. As radiocarbon data indicates that this island was also populated around the 3rd century, this bottleneck evidences a strong population size reduction after the initial colonization of El Hierro.

## Discussion

We analyzed genome-wide data from 49 individuals from the CIP, including 40 newly generated samples and 9 whole-genome sequencing data from previously published studies[15,21]. With this larger dataset, we have shed light on the prehistory of North Africa. Data from the CIP, who colonized the archipelago around the first centuries CE, indicate that this North African population was composed of four main ancestral components. First, as observed for Late Neolithic Moroccans[5], the Canarian indigenous people have both North African Paleolithic/Early Neolithic and European Early Neolithic components. However, the contribution of the North African Paleolithic/Early Neolithic component is greater in the indigenous people than in the Kef El Baroud individuals, confirming that the impact of European Neolithic migrations was not homogenous in the North African region. In addition, the indigenous people show the presence of a steppe component, most probably associated with the migration of North Mediterranean populations into North Africa during the Bronze or the Iron Ages. Finally, we detect a small sub-Saharan African component implying the existence of trans-Saharan migrations in North Africa already before the first centuries CE, predating such gene flow inferred from modern DNA data[6,42].

Our dataset of the indigenous people of the Canary Islands has also allowed us to start understanding the challenges of human dispersals into oceanic islands and isolated environments. Based on the analysis of ROHs, all individuals show signs of a relatively small effective population size, likely due to the initial founder effect. Within the archipelago, we observe that insular populations are heterogenous regarding both their genetic composition and diversity. As observed for the mtDNA, the western and eastern islands are differentiated, with the islands closer to the continent having a greater affinity with prehistorical populations from Europe, while the western islands are more akin to prehistorical individuals from North Africa. In fact, qpAdm modeling determines that the contribution of these two components is slightly different in the two regions. When combined with radiocarbon data, these differences appear to have existed since the beginning of the indigenous colonization period and remained unchanged. It is worth mentioning that the islands closer to the continent exhibit a more diverse corpus of alphabetic inscriptions and variations in rock art compared to the western islands[43,44]. While records of the Libyco-Berber language are found throughout all the islands, Lanzarote and Fuerteventura have produced inscriptions belonging to the so-called Latino-Canarian alphabet that are not present elsewhere in the archipelago[43,44]. Another example of differences between regions is the presence of fig trees (*Ficus carica*) exclusively in Gran Canaria[45], suggesting that some differences in the cultural and biological background of the settlers were present since the start of the 1st millennium CE. However, it is important to note that, although our dataset spans from the 3rd to the 16th century CE, most of the individuals are dated after the 10th century and more data from the early colonization period would be needed to determine if populations differed in origin (the two regions being colonized by moderately different North African populations) or if asymmetrical migrations occurred at the beginning of the colonization phase.

Island sub-populations also show differences in their genome-wide genetic diversity. El Hierro, La Gomera, Lanzarote, and Fuerteventura islands show the effects of strong genetic drift resulting in the reduction of their effective population sizes over time. By the 10th–12th centuries CE, we observe that some individuals from these islands are the result of close-kin matings, as expected with the reduced population sizes at that time. These islands also exhibit low mtDNA diversities, with the partial or complete fixation of certain lineages, reinforcing the idea of strong genetic drift and the lack of gene flow. These results agree with these islands being isolated and inhabited by small populations. For the island of El Hierro, we detect a strong bottleneck around the 9th century, coinciding with the transition period between the Vandal Minimum and the Medieval Warm climate episodes[46]. However, radiocarbon dating indicates that this island was populated simultaneously with the rest of the archipelago[47], so this bottleneck would evidence a strong population size reduction long after the initial colonization of the island. Considering that El Hierro is an island with limited resources, temperature and rainfall changes during the 9th century could have heavily affected the availability of natural resources and crop production, potentially producing a severe bottleneck. More data from the other islands would be needed to determine if this phenomenon was also experienced by La Gomera, Lanzarote, and/or Fuerteventura. Gran Canaria, La Palma, and Tenerife present a completely different scenario. In these islands, genetic diversity is higher than in the other sub-populations, both inferred from the mtDNA and genome-wide data. Estimations of effective population sizes are also higher, and, for Tenerife and Gran Canaria, $N_e$ values remain constant over time. Furthermore, the detected founder effect coincides with the proposed colonization time, indicating that no additional bottlenecks occurred during the indigenous colonization period. These results evidence that these populations either had a comparably large size and could sustain genetic diversity over time or were not isolated, as proposed for Gran Canaria based on interpretations on the archaeological evidence[48]. However, IBD estimates and consistent differences in the ancestral contributions between Tenerife and Gran Canaria do not agree with having a significant gene flow between them. In this line, a constant $N_e$ value is against the idea of a significant migration event within the studied period. Differences in the genetic diversity of insular populations can be explained as the result of differences regarding the islands' size, their ecological diversity and their annual amount of rainfall. Gran Canaria, La Palma and Tenerife are medium size islands with a current average rainfall varying from c. 100 mm in the southern coasts to c. 800–1000 mm in the highest parts of the islands[49]. Those conditions provide higher availability of resources that could sustain larger and more diverse populations than in the smaller islands of El Hierro, La Gomera, and Lanzarote. Fuerteventura, despite having a larger size, currently receives c. 100 mm of rainfall per annum, being the most arid island of the archipelago[49]. Having into consideration the historical and archaeological records[50], it has been estimated that Tenerife, La Palma and Gran Canaria could have sustained a population of around 30,000–60,000 people at the beginning of the Castilian

                                                                          

conquest, while the remaining islands were probably populated by just 1000–3000 people.

Overall, this dataset has allowed us to better understand the prehistory of North Africa and to start producing a more detailed picture of the complex colonization process of the Canary Islands, where human resilience, isolation, and diverse insular environments led to the differentiation of their genetic landscape.

## Methods

### Ethics statement

Permissions needed to analyze ancient human remains were granted by the local authority (Dirección General de Patrimonio Cultural del Gobierno de Canarias; reference 51/2020-0717115014) and local museums. To assure ethical treatment of the archaeological remains and proper heritage conservation and dissemination, and facilitate the building of local capacities[51–53], (a) we only sampled the archaeological material strictly necessary to meet the objectives of this project, (b) we used less-destructive sampling methods such as the use of teeth and/or small bones, (c) we worked with local museums and institutions in order to secure the dissemination of our research to the general public in the Canary Islands, including science popularization talks, outreach activities for high school students and the inclusion of aDNA results in temporal and permanent exhibitions, and (d) we carried out this study involving international collaborations but led by a Canarian institution.

### Sampling, DNA extraction, library preparation, next-generation sequencing and whole-genome capture

Sample collection was carried out in collaboration with the universities of La Laguna (Tenerife) and Las Palmas de Gran Canaria (Gran Canaria); the insular museums of Gran Canaria (El Museo Canario), La Palma (Museo Arqueológico Benehaorita) and La Gomera (Museo Arqueológico de La Gomera), and Tibicena Arqueología y Patrimonio Ltd. (Gran Canaria). In total, 40 human remains from 22 archaeological sites were selected for this project (Fig. 1b). Information about the archaeological sites including available radiocarbon dates can be found in Supplementary Data 1 and Supplementary Note 1. All the analyses were carried out in an ancient DNA-dedicated area at University of California Santa Cruz (USA) and the Universidad de La Laguna (Spain). DNA extraction and library preparation were performed following ref. 5. Briefly, well-preserved teeth and bones were thoroughly decontaminated and pulverized using a mixer mill (MM Restch®). Bone or teeth powder was then extracted following a modified version of the protocol outlined in ref. 22, and DNA libraries were constructed following ref. 23. Ancient DNA libraries were then sequenced on an Ilumina's NextSeq 500 platform (paired-end reads, 2 × 75 bp and 2 × 42 bp). Samples with low endogenous DNA content (<10%) were also enriched in human endogenous DNA using whole-genome in-solution capture[54], while samples with high endogenous content (>10%) were additionally sequenced to saturation. To improve the coverage on informative regions, samples were also captured using baits covering the SNPs contained in the Illumina Multi-ethnic Genotyping Array (MEGA)[34].

### Radiocarbon dating

Radiocarbon dating was performed using Accelerator Mass Spectrometry (AMS) at Beta Analytic, Inc. (USA) and the Scottish Universities Environment Research Centre (SUERC) facilities. All individuals included in this study were radiocarbon dated, excluding those from which a date was already available (Supplementary Data 1). Bone and tooth collagen were sampled from the same specimen used for ancient DNA analysis, but enamel was also employed in samples where tooth collagen was exhausted in the DNA extraction. The [14]C dates were then calibrated with the internationally agreed IntCal20 atmospheric calibration curve using the OxCal online software version 4.4 (https://c14. arch.ox.ac.uk/oxcal.html)[55]. The two-sigma probability interval (95.4%), recommended by ref. 56, was used when discussing the [14]C measurements.

### Raw read processing

Reads were trimmed and filtered based on their quality score (BASEQ < 20) and read length (<30 bp), and adapters were removed using AdapterRemoval v. 2.1[57]. Paired-end reads were merged considering an 11 bp overlap and then aligned to the human reference genome build GRCh37 using BWA v. 0.7.12[58] with seed disabled. Low-quality (MAPQ < 25) and duplicated mapped reads, as well as reads with alternative mapping coordinates were removed using SAMtools v. 0.1.19[59]. Lastly, BAM files from different runs were merged for each sample using SAMtools merge. All individuals were assessed for damage patterns using MapDamage v. 2.0.2[60] (Supplementary Data 3.1), and ContamMix v. 1.0–10[61] and Schmutzi[62] used to calculate contamination rates. Molecular sex was determined using the ry estimate[63] (Supplementary Note 4.1). Individuals assigned as male were assessed for Y-chromosome haplogroups based on the SNPs form the ISOGG Y-DNA Haplogroup tree 2019–2020 database (v.15.73)[64] (Supplementary Note 4.2) and corroborated using pathPhynder's best path algorithm[65].

### Ancestry inference: PCA, ADMIXTURE, qpAdm and *f3*-statistics

Ancestry of the indigenous people was first inferred from a principal component analysis (PCA) using the MEGA-Human Genome Diversity Project (HGDP) reference panel[34], previously curated following ref. 5. We performed a PCA based on the MEGA-HGDP panel and then projected ancient individuals using both LASER and the lsqproject option from smartpca[66]. For LASER, the BAM files trimmed 3 bp at both ends were directly used for PCA using the default specifications. For lsqproject, we performed SNP calling on the ancient trimmed bam files using SAMtools mpileup and filtering out low-quality bases (BASEQ > 30). Pseudo-haploid calls were then obtained by randomly selecting one allele from the mpileup output. Ancient Canarians were also compared to the Human Origins panel using the Allen Ancient DNA Resource curated by the Reich Lab (https://reich.hms.harvard.edu/allen-ancient-dna-resource-aadr-downloadable-genotypes-present-day-and-ancient-dna-data, version 42.4)[25]. Previously published aDNA genomic data was also integrated into the Human Origins dataset (see Supplementary Note 6 for more information).

We used the same datasets for the unsupervised clustering analysis, as for the lsqproject PCA, but pruning for linkage disequilibrium using PLINK v1.90 (–indep-pairwise 200 25 0.4)[67]. The global ancestry of ancient people was determined using ADMIXTURE v1.3.0[26]. The analysis was performed in 10 replicates with different random seeds, and only the highest likelihood replicate for each value of *K* was taken into consideration. Alternatively, we performed an admixture modelling using qpAdm from ADMIXTOOLS following ref. 68. To account for sample size differences, we compared the qpAdm admixture values obtained at individual level by island (Supplementary Note 7). Finally, to determine the amount of shared drift between the CIP and other ancient populations we performed *f3*-statistics using ADMIXTOOLS[29]. For that purpose, we considered Ju/'hoansi North as the outgroup population and calculated the outgroup-*f3* value for all ancient populations contained in the Human Origins dataset (Supplementary Note 6).

### Diversity, family relationships and demographic history estimations

Diversity estimations were performed by calculating the whole-genome heterozygosity at a population and archaeological site level, and inbreeding patterns at the individual level. First, we estimated heterozygosity from each island and archaeological site using popstats[69]. We used the MEGA-HGDP panel and considered only archaeological sites with at least two individuals (Supplementary

Note 6.1). Alternatively, to evaluate the inbreeding patterns from the indigenous individuals, we assessed for the presence of runs of homozygosity (ROHs) using hapROH[36] with both the Human Origins and the MEGA-HGDP datasets. We screened for the total sum of ROH >4, >8, >12, >20 cM in individuals with genome coverage equal or higher than 0.30X. Furthermore, individuals from the same island were assessed for their kinship up to second-degree relationships using READ[70] with the MEGA-HGDP dataset (Supplementary Note 6.3). To evaluate changes in the effective population size ($N_e$) in the indigenous populations of the islands, we also used hapROH in combination with radiocarbon data. We used ASCEND[41] to infer the time and the intensity of putative bottleneck events that occurred in the archipelago (Supplementary Notes 9.1 and 9.2; Supplementary Data 9). For that purpose, we inferred bottlenecks in view of the geographical adscription of the individuals, considering (1) the archipelago as a whole, (2) eastern and western regions, and (3) each island, but only when at least five individuals were available. We corrected for sample bias in (1) and (2) by randomly choosing individuals based on the smallest sample size from each dataset. We also included El Hierro ($n = 4$) as all of the individuals have an average depth higher than 0.3X. For that, we also corrected for sample bias in the islands with $n > 5$ by performing 10 independent replicates using 4 individuals with similar characteristics to the sample from El Hierro. Finally, we inspected for shared genomic segments that are identical by descent (IBD) using IBDseq[40]. For that, we imputed genomes with an average depth >0.1X using GLIMPSE[71]. We first computed genotype likelihoods for each genome using the candidate variant sites contained in the 1000 Genomes Project phase 3[72], and filtering out for non-biallelic and singleton variants (details in Supplementary Note 12). We filtered out the individuals with an average genotype probability (GP) lower than 0.95 and removed variants with an INFO score <0.5. We then merged the Canarian individuals with a manually curated database with interesting ancient genomes and recalculated their INFO score filtering out variants with an INFO score <0.5. We called on the resulting VCFs the IBD segments using IBDseq[40] using the default parameters, and converted the IBD segments from base pair to centimorgan using the HapMap Phase II haplotype panel as in ref. 73. Finally, we carried out genetic clustering of the ancient individuals using hierarchical community detection on a network of pairwise IBD-sharing similarities, as in ref. 74. Briefly, we used igraph[75] to build a weighted network of the individuals using the fraction of the genome shared IBD between pairs as weights, and used this network to perform community detection using the Leiden algorithm[76] implemented in the leidenAlg R package (https://github.com/kharchenkolab/leidenAlg).

### Modern Canarian population

In order to compare the ancient population with the modern people of the Canary Islands, we first considered the Y-chromosome data and compared it to the modern DNA data from ref. 77. For that, we calculated admixture proportions using the mL estimator as in ref. 12, considering three main parental populations: the Canarian indigenous population, and the current Iberian and Sub-Saharan populations. Admixture estimator mL was calculated based on Y-chromosome haplogroup frequencies using the WLSAdmix program[78]. For the admixture proportions at genome-wide level, we built a dataset including the indigenous individuals, the Human Origins panel and the data generated by ref. 39. With that dataset, we applied qpAdm from ADMIXTOOLS as described for the ancient individuals, and considering the indigenous, Spanish and Yoruban populations as contributing sources.

### Reporting summary

Further information on research design is available in the Nature Portfolio Reporting Summary linked to this article.

## Data availability

The sequence data generated in this study have been deposited in the European Nucleotide Archive (ENA) database under accession number PRJEB61655. The rest of the indigenous Canarians included in this study were obtained from their ENA accession no. PRJEB86458[15] and PRJEB46005[21]. The human reference sequence build 37 (GRCh37/hg19 [https://www.ncbi.nlm.nih.gov/assembly/GCF_000001405.13/]) and the Revised Cambridge Reference Sequence (rCRS; NC_012920 [https://www.ncbi.nlm.nih.gov/nuccore/251831106]) were downloaded from the National Center for Biotechnology Information (NCBI). 1000 Genomes Project phase 3[72] used as reference dataset for imputation is available through the ENA accession number PRJEB31736. The Human Genome Diversity Project (HGDP) genotyped with the MEGA array[35] is available at https://bustamantelab.stanford.edu, and the ISOGG Y-DNA Haplogroup tree 2019–2020 database (v.15.73)[64] at https://isogg.org/tree/. The Allen Ancient DNA Resource dataset (AADR) version 42.4 is publicly available at https://reich.hms.harvard.edu/ancient-genome-diversity-project. The remaining ancient genomic data not included in the AADR were collected from the ENA through their accession no.: Neolithic genomes from Iran (PRJEB13189)[79]; ancient Mediterranean genomes (PRJEB35980)[80]; Bronze Age individuals from Greece (PRJEB37782)[81]; ancient Sardinians (PRJEB35094)[30]; Etruscans (PRJEB42866)[82]; and ancient genomes from the Iberian Peninsula (PRJEB46907)[83]; and from Great Britain (PRJEB47891)[84]. Array data from present-day Canary Islanders was obtained from[39] (https://www.iter.es/wp-content/uploads/2018/09/AffyCEU1_data_from_Canary_Islanders_MBE-Guillen-Guio-et-al.2018.zip).

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

## Acknowledgements

We want to acknowledge technical assistance from Nuria Álvarez-Rodríguez, Andre E. R. Soares, Joshua Kapp, Alexandra Sockell-Adams and Shirley Sutton.

## Author contributions

Conceptualization: R.F., J.S.C., M.A., A.R.R., J.M., C.D.B. Supervision: R.F., J.S.C., M.A., A.R.R., J.M., C.D.B. Selection of archaeological samples: A.C.O., J.S.C., E.S.C., M.A., A.R.R., J.M., J.V.V., V.A., T.D.D., M.C.C.M., J.C.H., M.A.M.B., J.P. Archaeological and anthropological contextualization: A.C.O., J.S.C., E.S.C., M.A., A.R.R., J.M., J.V.V., V.A., T.D.D., M.C.C.M., J.C.H., M.A.M.B., J.P. Laboratory work: R.F., A.C.O., J.G.S. Data analysis: J.G.S., R.F., A.C.O., J.S.C., E.S.C., H.R., M.S., H.M., M.P.Y., M.H.F. Funding acquisition: R.F., J.S.C., M.A., A.R.R., J.M., C.D.B. Writing—original draft: J.G.S., R.F., A.C.O. Writing—review & editing: J.G.S., A.C.O., J.S.C., E.S.C., M.A., A.R.R., J.M., J.V.V., V.A., T.D.D., M.C.C.M., J.C.H., M.A.M.B., J.P., H.R., M.S., H.M., M.P.Y., M.H.F., C.D.B., R.F.

## Competing interests

The authors declare no competing interests.
