## [Peer Review File · Nature Communications]

The genomic history of the indigenous people of the Canary IslandsREVIEWER COMMENTS

Reviewer #1 (Remarks to the Author):

This is a compelling paper that considers not only the population history of the Canary Islands, but the ways in which this history can inform about: 1) gaps in North African prehistory that are not currently addressed through aDNA; 2) island colonization histories and constraints on island populations, with broader applicability beyond the area of study.

The paper is solid from an archaeological point of view, with sufficient contextualization (where available) of archaeological human remains in the supplement, and a clear radiocarbon chronology that, whilst biased toward the second millennium AD, does provide a long time transect spanning both millennia AD.

I have some comments that address archaeological/anthropological concerns, leaving concerns about the aDNA laboratory and analytical methods to other reviewers:

- The supplement and the figures (map and chronology bar) provide excellent overview information, but some edits could make these stronger. For example, in Fig 1c, the use of two different rows is not clear – groups of closer and more islands, one presumes, but this should be clear in caption. Also, the authors could explain in the caption why CIP does not have a physical location (are these the decontextualized remains alluded to earlier?). Finally it is worth stressing in the paper that the majority of sampled individuals do come from 2nd millennium contexts (post-1200 for the most part) and more could be learnt from sampling earlier periods.
- More humanizing language could be used throughout the text and supplement, for example by changing “samples” to individuals, human remains, or people where appropriate. I also wonder about the absence of an ethics statement in the supplement or text. Since these remains seem to come from a number of different types of collections, both private and public (museum and university), with different contextual information, how was sampling conducted in a way that adheres to the ethical standards of these institutions, the professional community, and descendant communities?
- Some archaeological statements are underdeveloped and need references, such as this one: “Given the tremendous biogeographic diversity that characterizes the archipelago, the first settlers were driven to develop different life strategies that resulted in different adaptive processes in terms of social complexity, subsistence practices, and demographic development, making the settlement of the Canaries an extremely intriguing human colonization process.”
- On page 6, line 10, some more information is needed re: “decontextualized” human remains (this becomes clear but only much later in the paper).
- I know that space is limited in the discussion, but it would be nice to see a more robust discussion of the different factors that might be producing the differences observed between Tenerife, Gran Canaria, and La Palma on the one hand, and the other islands on the other. It is a bit confusing in the manuscript as to whether the authors see population size and genetic diversity as being linked to island size, resources, distance from mainland Africa and other islands, or a combination thereof, and here is where I think greater input from the archaeologist coauthors could be brought to bear on the discussion, since of course this interpretation hinges upon prior research.

Reviewer #2 (Remarks to the Author):

In this paper Serrano et al. generate and analyze ancient DNA from the Canary Island archipelago with the overall goal of understanding the pre-history of North Africa, primarily before the Arab conquest of the region. The authors generated paleogenomic data from all seven main islands from 40 new teeth and bone samples (a mix of whole genome and two different captures) dating from the 3rd

(at almost the beginning of its colonization based on archaeology) to 16th century CE. Amongst other things they demonstrate that these Canary Island populations are most related to Late Neolithic North Africans, but with additional European admixture, as well as clear evidence of population structure between the western and eastern island (the eastern populations appear to have more European ancestry). There does not appear to be much variation within islands for these ancestry proportions. Certain islands appear to have lower diversity and more runs of homozygosity (though all islands are enriched for the latter), and this reduced diversity has increased over time in those specific low diversity islands. The islands were likely quite insular based on ROH and IBD analysis, with very limited geneflow between them. Finally the authors find evidence for a series of bottlenecks, consistent with island founder effects, both for the whole archipelago and the western and eastern regions (El Hierro experiencing a particular strong bottleneck recently).

Overall this paper was a pleasure to read, and I do not have much in the way of revisions. The paper is well written, of appropriate length, and its conclusions are fully justified by the results. The data is generated appropriately, and authors use a variety of population genetic methods to entangle the prehistory of the region at multiple scales, revealing very fine detail of the demography of the archipelago over the last ~1000 years. The authors should be particularly applauded for the care in their analysis and how they then interpret the results. The conclusions are based on robust and comprehensive analyses that often aim to mitigate any potential issues such as differences in coverage or sample size that could bias their interpretation. I believe this results of this paper are robust and will be of great interest to those working in this region across disciplines and amongst the population genetics community. I have no suggested major (and very few minor revisions)

Minor Comments.

The results based on comparison to the MEGA-HGDP dataset seem a little redundant and underpowered, as most of the resolution is seen with the Human Origins panel (as the authors admit). The paper may be a little bit easier to read by excluding the former, rather than having to keep referring to the supplement to see the MEGA-HGDP results (which do not add anything as far as I can tell).

Y-chromosome and mtDNA results seems a little over-emphasized, particularly in the supplement.

Some data from mainland North Africa from the same period would have been interesting to understand the proposed steppe-like component not seen in the Late Neolithic Moroccans, but this is not a criticism of the study, as obviously such data is hard to find and generate.

Supplementary Fig. 16. Maybe a zoom in would be helpful, I cannot see much with regard to where the replicates are here in these plots.

Supplementary Fig. 20. The complete yellow ancestry of western islands seems weird here, especially given the results from the Human Origins panel. I think this likely relates to my above comment that the HGDP data is underpowered for this analysis compared to the Human Origins data.

RESPONSE TO THE REVIEWERS:

Reviewer 1:

First of all, we want to thank reviewer 1 for the time devoted to review our manuscript and for her/his helpful comments.

Below, we provide point-by-point answers to all your concerns:

Point 1 – We agree with the reviewer that Figure 1 can be improved. We added in Fig. 1C “Eastern Islands” and “Western Islands” to each row of radiocarbon-dated individuals in order to clarify why they are placed in such a way (page 35). Regarding the individuals marked as “CIP”, in the new version we separated in Fig. 1a published samples from the Canary Islands (previously marked as “CIP”) based on their original publication and the location of their archaeological site, when available. The individuals sampled by Rodriguez-Varela et al. (2017) have no information available on their archaeological site to correctly locate them, so we cannot include them in the archaeological site map from Fig. 1b. All this information is now included in the legend of Figure 1 (page 35, lines 4 – 11). Regarding the bias toward later radiocarbon dates, we agree that including individuals from the earlier colonization period is of paramount importance. However, the smaller sample size from the first centuries of the indigenous period is not a product of our experimental design but a reflection of lower availability of archaeological sites with those chronologies, being more common those dated after the 10th century. In the new version, we have acknowledged this limitation (page 18, lines 8 – 12).

Point 2 – We strongly agree with the reviewer that a more humanizing language should be used in scientific publications studying human remains and that ethic statements should be included in the main text. We have changed “samples” to “individuals”, “human remains” or “people” through the manuscript and supplementary information when appropriate. The ethic statement, previously included in the cover letter, is now in the Methods section (page 20, lines 15 – 21). Briefly, permission was granted for the analysis of human remains from both the local museums and the local authority (Dirección General de Patrimonio Cultural del

Gobierno de Canarias; reference 51/2020-0717115014). In order to adhere to recommendations for the analysis of human remains, we only sampled the archaeological material strictly necessary to meet the objectives of this project, we used less-destructive sampling methods such as the use of teeth and/or small bones, and we worked with local museums and institutions to in order to secure the dissemination of our research to the general public in the Canary Islands. In this last item, as this research has been led by Canarian researchers at a Canarian institution, we have had ample opportunity to disseminate the results of the study in real time through numerous outreach activities. These activities have included visits to schools and high schools to show them how we study their ancestors, talks for the general public to keep them informed on the project results, participation in interviews and documentaries, and collaborating with local museums to produce graphical material detailing our results for their exhibition explaining what ancient DNA tells us about our past.

Point 3 – As requested by reviewer 1, several references have been added to support the paragraph beginning with “Given the tremendous biogeographic diversity that characterizes the archipelago (...)” (page 6, line 3), including evidence for temporal dietary changes within the archipelago or the abandonment over time of certain agricultural practices:

- Sánchez-Cañadillas, E. et al. Dietary changes across time: Studying the indigenous period of La Gomera using $\delta^{13}\text{C}$ and $\delta^{15}\text{N}$ stable isotope analysis and radiocarbon dating. *Am. J. Phys. Anthropol.* 175, 137–155 (2021).
- Morales, J., Rodríguez, A., Alberto, V., Machado, C. & Criado, C. The impact of human activities on the natural environment of the Canary Islands (Spain) during the pre-Hispanic stage (3rd–2nd Century BC to 15th Century AD): an overview. *Environ. Archaeol.* 14, 27–36 (2009).
- Mitchell, P. J. Archaeological Research in the Canary Islands: Island Archaeology off Africa’s Atlantic Coast. *J. Archaeol. Res.* (2023).
- Martín Rodríguez, E. Adaptación y adaptabilidad de las poblaciones prehistorias canarias: una primera aproximación. *Vegueta Anu. Fac. Geogr. E Hist.* 9–19 (1993).

Point 4 – In page 6 paragraph 2 (lines 11 - 12), we have specified why the previously published individuals from the Canary Islands (Rodriguez-Varela et al. 2017) are considered “decontextualized”.

Point 5 – In page 18 (lines 1 – 7), page 19 (lines 18 – 23) and page 20 (lines 1 – 7), we added possible archaeological and environmental explanations for the differences observed on the genetic composition and the genetic diversity within insular populations. Regarding differences on their genetic composition, we discuss differences observed in inscriptions found in Lanzarote and Fuerteventura, compared to the remaining islands, with a particular alphabet observed only in the two islands closer to the continent. We also present recent archaeobotanical evidence on the presence of particular domestic plants only in the island of Gran Canaria, pointing to possible cultural differences since the beginning of the colonization process. However, we conclude that more evidence from the earlier colonization is needed to understand what is the more probable explanation for the differences observed between eastern and western islands. Regarding diversity values, we have included information on the different environmental conditions of islands showing higher and lower diversity values, and population size estimates from archaeological and historical evidence. As inferred from effective population size estimates, archaeological and historical evidence pointed to larger human populations in La Palma, Tenerife and Gran Canaria (30,000 – 60,000 people) compared to El Hierro, La Gomera, Lanzarote and Fuerteventura (1,000 – 3,000 people).

Reviewer 2:

We want to thank reviewer 2 for the time devoted to review our manuscript and for her/his helpful comments.

Below, we provide point-by-point answers to all your concerns:

Point 1 – We agree with the reviewer that the use of the MEGA-HGDP reference could be seen as redundant. In the new version, we have excluded PCA and ADMIXTURE results using MEGA-HGDP from the main text (page 7, lines 11 – 13).

However, as this was the panel used for targeted capture, we used it for analyses that were benefited by a higher coverage, such as the heterozygosity estimation and the identification of ROHs. What panel we used for each analysis is indicated in the main text and explained in detail in the Supplementary File.

Point 2 – We understand that our discussion of mitochondrial DNA and Y-chromosome results in the Supplementary Material document may seem over-emphasized. However, it is important to note that most of the previous genetic studies of the ancient people of the Canary Islands have been focused on the analysis of these uniparental markers. Therefore, we believe that a comprehensive comparison of our mtDNA and Y-chromosome results with the ones from the previous literature is indispensable to properly contextualize our study within existing molecular evidence. Maybe the reviewer does not agree with us, but our experience is that, sometimes, previous mtDNA and Y-chromosome analyses are not properly taken into consideration in paleogenomic studies, and in rare occasions they are even not considered at all. We just wanted to properly acknowledge previous evidence in our study population and integrated it into our discussion.

Point 3 – As requested, we have modified Supplementary Fig. 16 by zooming in on the PCA replicates of the indigenous individuals (page 50).

Point 4 – We acknowledge the concerns raised by the reviewer regarding the power of the ADMIXTURE analysis using MEGA-HGDP when compared to those using the Human Origins panel. For that reason, we have now removed all reference to this analysis in the main text (page 7, lines 20 – 22). However, we believe that the inclusion of Supplementary Fig. 20 is necessary to illustrate section 5.3.1 in the Supplementary Material. Since the Human Origins capture kit was not publicly accessible when our samples were processed, we captured the individuals using the MEGA-HGDP. For that reason, we think it is relevant to perform all the analyses using this panel, even if they are limited, and include them in the Supplementary File. Now, we have modified the text (Supplementary File, page 19, lines 3 – 5) to better clarify that this dataset is not appropriate to resolve the genetic composition of the indigenous people of the Canary Islands in fine detail.

We have included the European Nucleotide Archive database project number “PRJEB61655” to the "Data and materials availability" section, indicating where the sequencing data presented in this paper will be publicly available (page 27, line 1).

REVIEWERS' COMMENTS

Reviewer #1 (Remarks to the Author):

I have reviewed the revised manuscript and the responses to reviewer comments and I find that the comments I provided have been satisfactorily addressed. The new manuscript is stronger as a result. I recommend its publication.